# DUALITY OF INFORMATION FLOW: INSIGHTS IN GRAPHICAL MODELS AND NEURAL NETWORKS

## ABSTRACT

This research highlights the convergence of probabilistic graphical models and neural networks, shedding light on their inherent similarities and interactions. By interpreting Bayesian neural networks within the framework of Markov random fields, we uncovered deep connections between message passing and neural network propagation. Our exploration unveiled a striking equivalence between gradients in neural networks and posterior-prior differences in graphical models. Empirical evaluations across diverse scenarios and datasets showcased the efficacy and generalizability of our approach. This work introduces a novel perspective on Bayesian Neural Networks and probabilistic graphical models, offering insights that could pave the way for enhanced models and a deeper understanding of their relationship.

## 1    INTRODUCTION

Probabilistic graphical models and neural networks are two distinct paradigms for modeling data generation within networks composed of fundamental computational units. A probabilistic graphical model defines the joint probability distribution of a network of random variables by leveraging conditional probabilities or clique potentials. In contrast, a neural network characterizes the transformation of a tensor "particle" as it progresses through multiple layers, encompassing both linear and nonlinear operations, to achieve specific outcomes defined by a loss function. In a broader context, each computational graph, trained with a dataset sampled from a data distribution via stochastic gradient descent, gives rise to a probabilistic graphical model. This model delineates a joint probability distribution involving synaptic weights and the data. Conversely, every probabilistic graphical model leads to a computational graph in which stochastic message passing strives to attain detailed balance, ultimately resulting in a stationary joint probability distribution. While simulation-based approaches are often more scalable for processing extensive datasets when compared to analytical methods, the exploration of connections between deterministic and stochastic perspectives within network-based computational models holds the promise of yielding enhanced models and deeper insights (Bishop, 2006; Goodfellow et al., 2016).

In our research, we conceptualized a Bayesian neural network as a Markov random field (Neal, 1995), where the loss function serves as potential energy governing the trajectories of tensors as they navigate through the computational graph and interact with synaptic weights. The focus was on identifying mean parameters of tensors during forward propagation and computing gradients with respect to their canonical parameters in backward propagation (Rockafellar, 1970; Wainwright & Jordan, 2008; Khan & Rue, 2021). This process reveals the gradient as the difference in mean parameters between the posterior and prior distributions, highlighting a parallel between probabilistic graphical models' message-passing and neural networks' forward/backward propagation. Furthermore, we formulated the mean parameters during forward propagation and their sensitivities during backward propagation in terms of the statistics of tensor particles as they advanced through the computational graph and the gradients of loss with respect to these tensors as they propagated backward within the graph. This approach sheds light on the duality between probabilistic graphical models and Bayesian neural networks, linking population dynamics and distributions. In our paper, we use the term "particle" to describe a set comprising inputs, post-activations, synaptic weights, and labels that collectively constitute the state of a Bayesian neural network. This terminology is borrowed from Monte Carlo methods prevalent in statistical physics

and computational mathematics, where a particle represents a possible state of the system under study, analogous to how physical particles behave and interact (Gardiner et al., 1985).

Our algorithm underwent extensive testing across various datasets including CIFAR 10/100, Tiny ImageNet, and UCI regression, using neural network architectures like DenseNet, ResNet, and others. We explored different learning algorithms, such as Bayes-by-backprop for Bayesian neural networks (BNNs), and incorporated techniques like cosine learning rate scheduling and image augmentation. We also addressed vanishing gradient issues in BNNs using the variational message passing algorithm (Winn et al., 2005) and showcased BNNs' superior generalization through test error analysis and visualization of learned data distributions.

## 2 NOTATION

A neural network, as explained in Goodfellow et al. (2016), emulates decision-making processes based on a training dataset $D$ by minimizing a specific loss. This loss is essentially the empirical mean of the loss function $J(\hat{\mathbf{y}}(\mathbf{x}, \mathbf{W}), \mathbf{y})$, calculated in the following manner:

$$\text{argmin}_{W=\{\mathbf{W}_1 \dots, \mathbf{W}_L\}} \frac{1}{|D|} \sum_{(\mathbf{x}, \mathbf{y}) \in D} J\left(\hat{\mathbf{y}}(\mathbf{x}, \mathbf{W}), \mathbf{y}\right), \tag{1}$$

where $\hat{\mathbf{y}}(\mathbf{x}, \mathbf{W}) = \mathbf{x}_L, \mathbf{x}_l = \mathbf{f}_l(\mathbf{a}_l), \mathbf{a}_l = \mathbf{W}_l \cdot \mathbf{x}_{l-1} + \mathbf{b}_l$ for $l = 1, \dots, L$, and $\mathbf{x}_0 = \mathbf{x}$.

In these equations, $\mathbf{a}_l$, $\mathbf{x}_l$, $\mathbf{W}_l$, and $\mathbf{f}_l$ denote the pre-activation, post-activation, weights, and activation function of each layer $l$, respectively. The size of the dataset $|D|$ is the count of training examples. Often, the loss function $J$ is the negative log-likelihood, $J(\hat{\mathbf{y}}, \mathbf{y}) = -\log p(\mathbf{y}; \hat{\mathbf{y}})$, capturing the cross-entropy between the empirical distribution of training data and the probabilistic model. In our notation, operators are applied from the left, meaning $\mathbf{W}_l \cdot \mathbf{x}_{l-1} = \left(\sum_{j_{l-1}} \mathbf{W}_{i_l, j_{l-1}} \mathbf{x}_{j_{l-1}}\right)_{i_l}$ computes pre-activation elements at multi-index $i_l$ as the weighted sum of post-activation elements at multi-index $j_{l-1}$. For ease of understanding, multi-indices can be treated as standard integer indices, with synaptic weights represented as matrices and post-activation tensors as vectors. The dimensionality of these indices depends on the layer's architecture.

Let $\boldsymbol{\delta}_l = \nabla_{\mathbf{a}_l} J(\hat{\mathbf{y}}, \mathbf{y})$ denote the gradient of the loss with respect to the activation inputs at layer $l$, commonly referred to as the sensitivity of activation inputs (Stork et al., 2000). Backpropagation recursively computes these gradients for activation inputs and weights through automatic differentiation: $\boldsymbol{\delta}_L = \mathbf{f}'_L \circ \nabla_{\hat{\mathbf{y}}} J(\hat{\mathbf{y}}, \mathbf{y})$, $\boldsymbol{\delta}_{l-1} = \mathbf{f}'_{l-1} \circ \left(\mathbf{W}_l^\top \boldsymbol{\delta}_l\right)$, and $\nabla_{\mathbf{W}_l} J(\hat{\mathbf{y}}, \mathbf{y}) = \boldsymbol{\delta}_l \mathbf{x}_{l-1}^\top$, for $l$ ranging from $L$ to 1. Here, $\nabla$ represents the gradient, $\circ$ indicates element-wise multiplication, $\bullet^\top$ signifies matrix transpose, and $\mathbf{f}'_l$ represents the derivative of the activation function.

A Bayesian neural network (Neal, 1995) mimics decision-making behavior in training data using an ensemble of neural networks that share the same computational graph but have distinct synaptic weights. In this paper, we frame the learning problem as the minimization of the following variational principle over a set of variational posterior parameters $\boldsymbol{\theta}$:

$$\mathbf{E}_{q(\mathbf{W}; \boldsymbol{\theta}_\mathbf{W})} \sum_{(\mathbf{x}, \mathbf{y}) \in D} J\left(\hat{\mathbf{y}}(\mathbf{x}, \mathbf{W}), \mathbf{y}\right) + \log q(\mathbf{W}; \boldsymbol{\theta}_\mathbf{W})/p(\mathbf{W}). \tag{2}$$

Here, $\mathbf{W} = \{\mathbf{W}_1 \dots, \mathbf{W}_L\}$ represents the weights, $p(\mathbf{W})$ is the prior probability, and $q(\mathbf{W}; \boldsymbol{\theta}_\mathbf{W})$ is the variational posterior probability. All other symbols are consistent with those of a non-Bayesian neural network. The objective is to minimize the negative evidence lower bound. The stochastic weights introduce a probability distribution over $\hat{\mathbf{y}}(\mathbf{x})$. The ensemble decision is formulated as Bayesian model averaging $\mathbf{E}_{q(\mathbf{W}; \boldsymbol{\theta}_\mathbf{W})} \hat{\mathbf{y}}(\mathbf{x}|\mathbf{W})$, and model uncertainty can be assessed through the entropy of the ensemble decision $-\mathbf{E}_{q(\mathbf{W}; \boldsymbol{\theta}_\mathbf{W})} \log p(\hat{\mathbf{y}}|\mathbf{x}, \mathbf{W})$.

Minimizing the variational free energy in the aggregated loss $-\sum_{(\mathbf{x}, \mathbf{y}) \in D} \log p(\mathbf{y}|\mathbf{x})$ concurrently minimizes the variational free energy in the negative log-likelihood of the training data $-\sum_{(\mathbf{x}, \mathbf{y}) \in D} \log p(\mathbf{x}, \mathbf{y})$, because $\log p(\mathbf{x})$ is independent of $\mathbf{W}$. Therefore, optimizing Eq. 2 with respect to the variational parameters $\boldsymbol{\theta}_\mathbf{W}$ provides a sampling-free methodology (Tieleman & Hinton, 2009; Friston, 2010; Lee & LeCun, 2017; Song et al., 2021) for learning complex probability distributions over the data $D$. Here, the potential energy is defined as $\log p(\mathbf{x}, \mathbf{y}, \mathbf{W}) = J(\hat{\mathbf{y}}(\mathbf{x}, \mathbf{W}), \mathbf{y}) - \log p(\mathbf{W}) + \text{constant}$.

As tensor "particles" traverse multiple layers of linear and nonlinear transformations within an ensemble of neural networks during forward propagation, their probability distribution evolves accordingly. Likewise, during backpropagation, as the loss gradient with respect to these tensors traverses these transformation layers, it characterizes the sensitivity of the probability distributions. This sensitivity, in turn, generates various probability kernels that guide the particles and modify the probability distributions toward minimizing the loss. In this context, a duality (Gardiner et al., 1985) emerges between the stochastic tensor flow within a Bayesian neural network and the deterministic evolution of probability distributions within a probabilistic graphical model.

The case of particular interest is when the data-generating process from $\mathbf{x}_0$ to $\mathbf{x}_{l=1,\ldots,L}$ and $\mathbf{y}$ is approximately Gaussian-linear. As suggested by Jacot et al. (2018), neural networks tend to remain approximately linear throughout training in overparameterized regimes. This linear approximation simplifies the analysis of convergence and generalization, as it essentially involves inferences using multivariate normal distributions. In such scenarios, the activation function can be approximated using its first-order Taylor expansion. Additionally, the synaptic weights are reparameterized using weight parameters $\boldsymbol{\theta}_{\mathbf{W}_l}$ and a standard Gaussian random vector $\boldsymbol{\omega}_l$. This leads to a Gaussian Markov random field, as described by Rue & Held (2005), characterized by the following Langevin process (describing the stochastic tensor flow) and the Fokker-Planck process (detailing the tensor distribution evolution):

$$\mathbf{x}_l = \mathbf{f}_l(\mathbf{x}_{l-1}, \mathbf{W}_l) \approx \hat{\mathbf{x}}_l + \hat{\mathbf{f}}'_{\mathbf{x}_{l-1}} \cdot (\mathbf{x}_{l-1} - \hat{\mathbf{x}}_{l-1}) + \hat{\mathbf{f}}'_{\mathbf{W}_l} \cdot \left(\mathbf{W}_l - \hat{\mathbf{W}}_l\right),$$

$$\mathbf{y} = \mathbf{g}(\mathbf{x}_L, \boldsymbol{\nu}_L) \approx \mathbf{g}(\hat{\mathbf{x}}_L, 0) + \hat{\mathbf{g}}'_{\mathbf{x}_L} \cdot (\mathbf{x}_L - \hat{\mathbf{x}}_L) + \hat{\mathbf{g}}'_{\boldsymbol{\nu}_L} \boldsymbol{\nu}_L,$$

$$p(\mathbf{x}_l | \mathbf{x}_{l-1}; \boldsymbol{\theta}_{\mathbf{W}_l}) = \mathcal{N}\left(\hat{\mathbf{x}}_l + \hat{\mathbf{f}}'_{\mathbf{x}_{l-1}} \cdot (\mathbf{x}_{l-1} - \hat{\mathbf{x}}_{l-1}), \hat{\mathbf{f}}'_{\mathbf{W}_l} P_{\mathbf{W}_l} \hat{\mathbf{f}}'^{\top}_{\mathbf{W}_l}\right), \tag{3}$$

$$p(\mathbf{y} | \mathbf{x}_L) = \mathcal{N}\left(\mathbf{g}(\hat{\mathbf{x}}_L, 0) + \hat{\mathbf{g}}'_{\mathbf{x}_L} \cdot (\mathbf{x}_L - \hat{\mathbf{x}}_L), \mathbf{g}'_{\boldsymbol{\nu}_L} \mathbf{g}'^{\top}_{\boldsymbol{\nu}_L}\right). \tag{4}$$

In the above equations, the transformation $\mathbf{f}_l$, mapping post-activation $\mathbf{x}_{l-1}$ to post-activation $\mathbf{x}_l$, is approximated using a first-order Taylor expansion centered around $\mathbf{x}_{l-1} = \hat{\mathbf{x}}_{l-1}$ and $\hat{\mathbf{W}}_l$. Specifically, $\hat{\mathbf{x}}_l = \mathbf{f}_l(\hat{\mathbf{x}}_{l-1}, \hat{\mathbf{W}}_l)$ represents the mean transformation, and $\hat{\mathbf{f}}'_{\mathbf{x}_{l-1}}$ and $\hat{\mathbf{f}}'_{\mathbf{W}_l}$ are the gradients of $\mathbf{f}_l$ with respect to $\mathbf{x}_{l-1}$ and $\mathbf{W}_l$, computed at this point. Similarly, the transformation $\mathbf{g}$ that maps post-activation $\mathbf{x}_L$ and the multivariate standard Gaussian noise generator $\boldsymbol{\nu}_L$ to the observation $\mathbf{y}$ is also approximated using a first-order Taylor expansion centered around $\mathbf{x}_L = \hat{\mathbf{x}}_L$ and $\boldsymbol{\nu}_L = 0$.

Learning a Gaussian linear Bayesian neural network involves inferring the posterior distributions for post-activations and weights through variational or MCMC methods. This process includes a forward "filtering" pass, updating Bayesian beliefs $\alpha(\mathbf{x}_l; \boldsymbol{\theta}_l) \stackrel{\text{def}}{=} p(\mathbf{x}_l | \mathbf{x}_0 = \mathbf{x})$ from the input to higher-level post-activations and output, and a backward "smoothing" pass, refining the beliefs $\gamma(\mathbf{x}_l; \boldsymbol{\theta}_{l|L}) \stackrel{\text{def}}{=} p(\mathbf{x}_l | \mathbf{x}_0, \mathbf{y})$, and $q(\mathbf{W}_l; \boldsymbol{\theta}_{\mathbf{W}_l})$ using information from the output and higher-level post-activations, where $\boldsymbol{\theta}_l$, $\boldsymbol{\theta}_{l|L}$, and $\boldsymbol{\theta}_{\mathbf{W}_l}$ are variational parameters. The Kalman filter/smoother, a deterministic algorithm, computes the mean and variance of post-activations and weights as follows:

$$\alpha(\mathbf{x}_l) = \mathcal{N}(\hat{\mathbf{x}}_l, P_l), \text{with } P_l = \hat{\mathbf{f}}'_{\mathbf{x}_{l-1}} P_{l-1} \hat{\mathbf{f}}'^{\top}_{\mathbf{x}_{l-1}} + \hat{\mathbf{f}}'_{\mathbf{W}_l} P_{\mathbf{W}_l} \hat{\mathbf{f}}'^{\top}_{\mathbf{W}_l}, \tag{5}$$

$$\gamma(\mathbf{x}_l) = \mathcal{N}\left(\hat{\mathbf{x}}_{l|L}, P_{l|L}\right), \text{ with } \hat{\mathbf{x}}_{L|L} = \hat{\mathbf{x}}_L + K_L(\mathbf{y} - \hat{\mathbf{y}}), P_{L|L} = P_L - K_L S_L K_L, \tag{6}$$

$$\hat{\mathbf{x}}_{l|L} = \hat{\mathbf{x}}_l + G_l\left(\hat{\mathbf{x}}_{l+1|L} - \hat{\mathbf{x}}_{l+1}\right), P_{l|L} = P_l - G_l\left(P_{l+1|L} - P_{l+1}\right)G_l^{\top},$$

$$\hat{\mathbf{W}}_{l|L} = \hat{\mathbf{W}}_l + G_{\mathbf{W}_l}\left(\hat{\mathbf{x}}_{l|L} - \hat{\mathbf{x}}_l\right), P_{\mathbf{W}_{l|L}} = P_{\mathbf{W}_l} - G_{\mathbf{W}_l}\left(P_{l|L} - P_l\right)G_{\mathbf{W}_l}^{\top},$$

where $S_L = \hat{\mathbf{g}}'_{\mathbf{x}_L} P_L \hat{\mathbf{g}}'^{\top}_{\mathbf{x}_L} + \hat{\mathbf{g}}'_{\boldsymbol{\nu}_L} \hat{\mathbf{g}}'^{\top}_{\boldsymbol{\nu}_L}$ represents the observation covariance, $K_L = P_L \mathbf{g}'^{\top}_{\mathbf{x}_L} S_L^{-1}$ is the Kalman gain from the observation, $G_l = P_l \hat{\mathbf{f}}'^{\top}_{\mathbf{x}_l} P_{l+1|L}^{-1}$ and $G_{\mathbf{W}_l} = P_{\mathbf{W}_l} \hat{\mathbf{f}}'^{\top}_{\mathbf{W}_l} P_{l|L}^{-1}$ denote the smoothing gains in backpropagation, and $\hat{\mathbf{W}}_{l|L}$, $P_{\mathbf{W}_{l|L}}$, $\hat{\mathbf{W}}_l$, and $P_{\mathbf{W}_l}$ are the posterior and prior parameters of the weight $\mathbf{W}_l$.

## 3 MARKOV RANDOM FIELD SPECIFIED BY BAYESIAN NEURAL NETWORK

In this section, we consider tensor transformations within a Bayesian neural network as a stochastic process. We then formulate the forward propagation of tensor distributions and the

backpropagation of their sensitivities as a form of belief propagation. We establish a connection between the stochastic perspective on tensor transformations and the deterministic perspective on tensor distribution evolution. Lastly, we present an algorithm for converting a tensor layer into a stochastic layer.

## 3.1 BACKPROPAGATION THROUGH STOCHASTIC LAYERS

We will explore the forward propagation of tensor distribution parameters in a Bayesian neural network and the subsequent backward propagation of the loss gradient relative to these parameters within the network's computational graph. A theorem presented here establishes a profound mathematical link between two pivotal concepts: backpropagation, essential in neural network training, and belief propagation, commonly employed in probabilistic graphical models. This theorem sheds light on the flow of information through a Bayesian neural network during both forward and backward passes.

In its essential form, the filtering distribution $\alpha(\mathbf{x}_l; \boldsymbol{\theta}_l) = p(\mathbf{x}_l|\mathbf{x}_0 = \mathbf{x})$ is parameterized by $\boldsymbol{\theta}_l$, while the smoothing distribution is $\gamma(\mathbf{x}_l) = p(\mathbf{x}_l|\mathbf{x}_0, \mathbf{y})$. For the exponential family, $\alpha(\mathbf{x}_l; \boldsymbol{\theta}_l) = \exp(-\boldsymbol{\theta}_l \cdot \mathbf{T}(\mathbf{x}_l) - A(\boldsymbol{\theta}_l))$, with canonical parameters $\boldsymbol{\theta}_l$, feature statistics $\mathbf{T}(\mathbf{x}_l)$, and mean parameters $\boldsymbol{\mu}_l = \mathbf{E}_{\alpha(\mathbf{x}_l)} \mathbf{T}(\mathbf{x}_l)$ and $\boldsymbol{\mu}_{l|L} = \mathbf{E}_{\gamma(\mathbf{x}_l)} \mathbf{T}(\mathbf{x}_l)$. In a Gaussian linear model approximation of a Bayesian neural network, post-activation mean and variance are $\hat{\mathbf{x}}_l$, $P_l$, $\hat{\mathbf{x}}_{l|L}$, and $P_{l|L}$, with canonical parameters $\boldsymbol{\eta}_l = P_l^{-1}\mathbf{x}_l$, $\boldsymbol{\Lambda}_l = -.5 \cdot P_l^{-1}$ for filtering, and $\boldsymbol{\eta}_{l|L}$, $\boldsymbol{\Lambda}_{l|L}$ for smoothing.

**Theorem 1.** *Equivalence between Backpropagation and Belief Propagation in a Bayesian Neural Network.*

*(1) Gradient of Loss with Respect to Parameters: In its most general form, the gradient of loss with respect to the post-activation filtering distribution parameters $\boldsymbol{\theta}_l$ and the variational weights distribution parameters $\boldsymbol{\theta}_{\mathbf{W}_l}$ can be expressed as:*

$$\nabla_{\boldsymbol{\theta}_l} J = -\mathbf{E}_{\gamma(\mathbf{x}_l)} \nabla_{\boldsymbol{\theta}_l} \log \alpha(\mathbf{x}_l; \boldsymbol{\theta}_l), \qquad \nabla_{\boldsymbol{\theta}_{\mathbf{W}_l}} J = -\mathbf{E}_{\gamma(\mathbf{x}_{l-1}, \mathbf{x}_l)} \nabla_{\boldsymbol{\theta}_{\mathbf{W}_l}} \log p(\mathbf{x}_l|\mathbf{x}_{l-1}; \boldsymbol{\theta}_{\mathbf{W}_l}).$$

*(2) Error Gradient for Canonical Parameters: When filtering distribution is in the exponential family, error gradient for the canonical parameters is the difference in the mean parameters between smoothing and filtering distributions. Backpropagation mirrors backward belief propagation:*

$$\nabla_{\boldsymbol{\theta}_l} J = \boldsymbol{\mu}_{l|L} - \boldsymbol{\mu}_l, \;\; \nabla_{\boldsymbol{\theta}_l} J = \nabla_{\boldsymbol{\theta}_l} \boldsymbol{\mu}_l \, \nabla_{\boldsymbol{\mu}_l} \boldsymbol{\mu}_{l+1} \, \nabla_{\boldsymbol{\mu}_{l+1}} \boldsymbol{\theta}_{l+1} \, \nabla_{\boldsymbol{\theta}_{l+1}} J, \;\; \nabla_{\boldsymbol{\theta}_{\mathbf{W}_l}} J = \nabla_{\boldsymbol{\theta}_{\mathbf{W}_l}} \boldsymbol{\mu}_l \, \nabla_{\boldsymbol{\mu}_l} \boldsymbol{\theta}_l \, \nabla_{\boldsymbol{\theta}_l} J.$$

*(3) Error Gradient for Post-Activation Mean and Variance: In a Gaussian linear Bayesian neural network, the error gradient for post-activation mean and variance is related to the difference between smoothing and filtering mean and variance in the corresponding Gaussian linear dynamics. Gradient backpropagation parallels belief backward propagation induced by smoothing gain $G_l$ and $G_{\mathbf{W}_l}$:*

$$\nabla_{\hat{\mathbf{x}}_l} J = -P_l^{-1} \left( \hat{\mathbf{x}}_{l|L} - \hat{\mathbf{x}}_l \right), \nabla_{P_l} J = -.5 P_l^{-1} \left( P_{l|L} - P_l + (\hat{\mathbf{x}}_{l|L} - \hat{\mathbf{x}}_l)(\hat{\mathbf{x}}_{l|L} - \hat{\mathbf{x}}_l)^\top \right) P_l^{-1}, \quad (7)$$

$$\nabla_{\hat{\mathbf{x}}_l} J = P_l^{-1} G_l P_{l+1} \cdot \left( \nabla_{\hat{\mathbf{x}}_{l+1}} J \right), \nabla_{P_l} J = P_l^{-1} G_l P_{l+1} \cdot \left( \nabla_{P_{l+1}} J \right) \cdot P_{l+1} G_l P_l^{-1},$$

$$\nabla_{\hat{\mathbf{W}}_l} J = P_{\mathbf{W}_l}^{-1} G_{\mathbf{W}_l} P_l \cdot \left( \nabla_{\hat{\mathbf{x}}_l} J \right), \nabla_{\boldsymbol{\sigma}_{\mathbf{W}_l}^2} J = \mathrm{diag} \left( G_{\mathbf{W}_l} P_l \left( \nabla_{P_l} J \right) P_l G_{\mathbf{W}_l} \right) \big/ \boldsymbol{\sigma}_{\mathbf{W}_l}^4. \quad (8)$$

*The error gradient for the canonical parameters of post-activation filter distributions in a Gaussian linear Bayesian neural network is the difference in the first and second moments between smoothing and filtering distributions in the corresponding Gaussian linear dynamics. Gradient backpropagation again parallels belief backward propagation induced by smoothing gain:*

$$\nabla_{\boldsymbol{\eta}_l} J = -\left( \hat{\mathbf{x}}_{l|L} - \hat{\mathbf{x}}_l \right), \qquad \nabla_{\boldsymbol{\Lambda}_l} J = -\left( P_{l|L} + \hat{\mathbf{x}}_{l|L} \hat{\mathbf{x}}_{l|L}^\top - P_l - \hat{\mathbf{x}}_l \hat{\mathbf{x}}_l^\top \right),$$

$$\nabla_{\boldsymbol{\eta}_l} J = G_l \cdot \nabla_{\boldsymbol{\eta}_{l+1}} J, \qquad \nabla_{\boldsymbol{\Lambda}_l} J = G_l \nabla_{\boldsymbol{\Lambda}_{l+1}} J \, G_l^\top - \hat{\mathbf{x}}_l \nabla_{\boldsymbol{\eta}_{l+1}}^\top J \, G_l^\top - G_l \nabla_{\boldsymbol{\eta}_{l+1}} J \, \hat{\mathbf{x}}_l^\top.$$

In the above, (1) is established through a common technique involving the manipulation of logarithms within gradients. It allows us to compute the gradient of loss with respect to the parameters of the filtering distribution.

$$\nabla_{\boldsymbol{\theta}_l} \log p(y|x_0) = \frac{\nabla_{\boldsymbol{\theta}_l} \int dx_l p(y|x_l, x_0) p(x_l|x_0; \theta_l)}{p(y|x_0)} = \int dx_l \frac{p(y|x_l, x_0) p(x_l|x_0)}{p(y|x_0)} \nabla_{\boldsymbol{\theta}_l} \log p(x_l|x_0; \theta_l).$$

The exponential family assumption in (2) generally holds as long as the joint probability density of the random variables is strictly positive, as per the Hammersley-Clifford theorem (Hammersley & Clifford, 1971). This assumption enables general forward-propagation of mean parameters and backpropagation of canonical parameter sensitivities. The gradients of weight variational posterior parameters can be computed using these mean parameters. In (3), sensitivity over mean and variance is converted into canonical parameter sensitivity, backpropagated using the probability kernel, and then reconverted into mean and variance sensitivity, as illustrated by expressions like $P_l^{-1} G_l P_{l+1} \cdot (\nabla_{\hat{\mathbf{x}}_{l+1}} J)$ and $P_l^{-1} G_l P_{l+1} \cdot (\nabla_{P_{l+1}} J) \cdot P_{l+1} G_l P_l^{-1}$. Both forward and backward propagation processes involve the exchange of "innovations" between target and current distribution mean parameters, represented as cross-entropy loss gradients in terms of the probability kernels. Equation 8 is presented in a differential form due to its dependence on weight parameterization. A detailed derivation of the natural gradient for Gaussian linear Bayesian neural networks is provided in Section D.

The proof is presented in Section A. In Section C, we derive the backpropagation formula for computing the gradient of the loss with respect to the filter distribution parameters in both a hidden Markov model and a Gaussian linear model. This derivation allows us to draw comparisons between backpropagation and backward message passing. Consequently, Theorem 1 serves a dual role: it enables variational inferences within the deep learning framework for probabilistic graphical models and contributes to the advancement of probabilistic graphical model techniques for Bayesian deep learning.

## 3.2 DUALITY BETWEEN BAYESIAN NEURAL NETWORK AND PROBABILISTIC GRAPHICAL MODELS

A Bayesian neural network represents a probabilistic ensemble of neural networks operating within a common computational graph. These networks guide tensor "particles" through this graph, adjusting their paths based on gradient information to minimize their respective losses. As they learn and adapt, the probability distribution describing their trajectories evolves. The subsequent theorem establishes a connection between the probabilistic perspective of tensor flow and the deterministic perspective of tensor distribution evolution, focusing specifically on the first and second moments of the gradient.

**Theorem 2.** *Relationship between the Langevin and the Fokker-Planck Dynamics of a Bayesian Neural Network.*

*(1) For post-activations distributed as per an exponential family, the error gradient relative to post-activation equals the potential energy gradient, resulting from differences in canonical parameters between filtering and smoothing distributions:*

$$\nabla_{\mathbf{x}_l} \log p(\mathbf{y}|\mathbf{x}_l, \mathbf{x}_0 = \mathbf{x}) = \nabla_{\mathbf{x}_l} \log \frac{p(\mathbf{x}_l|\mathbf{y}, \mathbf{x}_0 = \mathbf{x})}{p(\mathbf{x}_l|\mathbf{x}_0 = \mathbf{x})} = (\nabla_{\mathbf{x}_l} \mathbf{T}(\mathbf{x}_l))^\top (\boldsymbol{\theta}_l - \boldsymbol{\theta}_{l|L}).$$

*This difference in canonical parameters defines the orthogonal projection from the Jacobian of the sufficient statistics to the error gradient relative to post-activation. Here, $\bullet^+$ is the pseudo-inverse.*

$$\boldsymbol{\theta}_l - \boldsymbol{\theta}_{l|L} = (\mathbf{E}_{\mathbf{x}_l} \nabla_{\mathbf{x}_l} \mathbf{T}(\mathbf{x}_l))^+ (\mathbf{E}_{\mathbf{x}_l} \nabla_{\mathbf{x}_l} \log p(\mathbf{y}|\mathbf{x}_l, \mathbf{x}_0 = \mathbf{x})).$$

*(2) In Gaussian-linear Bayesian neural networks, drift and diffusion processes, defined by the error gradient and Hessian, guide post-activations toward the variational posterior of Gaussian linear dynamics as follows:*

$$\nabla_{\mathbf{x}_l} \log p(\mathbf{y}|\mathbf{x}_l, \mathbf{x}_0 = \mathbf{x}) = P_l^{-1}(\mathbf{x}_l - \hat{\mathbf{x}}_l) - P_{l|L}^{-1}(\mathbf{x}_l - \hat{\mathbf{x}}_{l|L}),$$

$$\nabla_{\mathbf{x}_l \mathbf{x}_l^\top} \log p(\mathbf{y}|\mathbf{x}_l, \mathbf{x}_0 = \mathbf{x}) = P_l^{-1} - P_{l|L}^{-1},$$

$$\nabla_{\hat{\mathbf{x}}_l} \log p(\mathbf{y}|\mathbf{x}_0 = \mathbf{x}) = \mathbf{E}_{\gamma(\mathbf{x}_l)} \nabla_{\mathbf{x}_l} \log p(\mathbf{y}|\mathbf{x}_l, \mathbf{x}_0 = \mathbf{x}),$$

$$\nabla_{P_l} \log p(\mathbf{y}|\mathbf{x}_0 = \mathbf{x}) = \nabla_{\mathbf{x}_l \mathbf{x}_l^\top} \log p(\mathbf{y}|\mathbf{x}_l, \mathbf{x}_0 = \mathbf{x}) + \mathbf{E}_{\gamma(\mathbf{x}_l)} (\nabla_{\mathbf{x}_l} \log p(\mathbf{y}|\mathbf{x}_l, \mathbf{x}_0 = \mathbf{x})) (\nabla_{\mathbf{x}_l} \log p(\mathbf{y}|\mathbf{x}_l, \mathbf{x}_0 = \mathbf{x}))^\top.$$

*(3) When the weight variance vanishes, making the state transition deterministic with $p(\mathbf{x}_{l+1}|\mathbf{x}_l) = \delta(\mathbf{x}_{l+1} - f(\mathbf{x}_l))$, Bayesian back propagation degenerates into non-Bayesian back propagation:*

$$\nabla_{\mathbf{x}_l} \log p(\mathbf{y}|\mathbf{x}_l) = \nabla_{\mathbf{x}_l} \mathbf{x}_{l+1} \nabla_{\mathbf{x}_{l+1}} \log p(\mathbf{y}|\mathbf{x}_{l+1}).$$

The proof is provided in Section B. In (1), we compute $\nabla_{\mathbf{x}_l} \log p(\mathbf{y}|\mathbf{x}_l, \mathbf{x}_0 = \mathbf{x})$ by introducing the filter probability distribution $\alpha(\mathbf{x}_l; \boldsymbol{\theta}_l)$. In (2), both $\nabla_{\mathbf{x}_l} J$ and $\nabla_{\mathbf{x}_l \mathbf{x}_l^\top} J$ describe the drift and diffusion of $\mathbf{x}_l$ toward its posterior distribution. In (3), we calculate $\nabla_{\mathbf{x}_l} \log p(\mathbf{y}|\mathbf{x}_l)$ by introducing the Dirac delta distribution for $\mathbf{x}_l$, which transfers the gradient onto its parameters, or by setting the prior variance $P_l$ of $\mathbf{x}_l$ to 0.

This theorem unites probabilistic tensor flow with deterministic tensor distribution evolution, showing that Langevin dynamics in Bayesian neural networks, particularly weight updates, are equivalent to the Fokker-Planck equation governing weight distribution evolution. This insight holds practical value for Bayesian deep learning, offering potential for novel optimization algorithms for improved generalization and convergence. It also deepens our understanding of how deep neural networks explore weight spaces during training.

As an illustrative example, approximating batch normalization as $\left(\frac{\mathbf{w}^\top \mathbf{x} - \boldsymbol{\mu}}{\boldsymbol{\sigma}} + V\right) U$ aligns with variational Bayesian learning using stochastic scale-bias distributions $q(U)$ and $q(V)$, with a uniform prior on $\mathbf{w}$ ensuring $\|\mathbf{w}\| = 1$ (Shekhovtsov & Flach, 2019). This approach hints at using Bayesian layers to streamline neural network designs by reducing dependency on normalization layers (Zagoruyko & Komodakis, 2017; Brock et al., 2021). Additionally, gated recurrent unit-based networks can be conceptualized as hierarchical graphical models, incorporating binary features for input selection (Garner & Tong, 2020), paving the way for novel Bayesian recurrent neural network designs and interpretations. Furthermore, treating Bayesian neural networks as Gaussian Markov random fields offers new algorithmic possibilities and insights in Bayesian optimization, architecture design, and generalization (Snoeyink & Picheny, 2012; Arora et al., 2019a;b).

### 3.3 A Deterministic BNN Backpropagation Algorithm

Given the dual relationship between forward-backward propagation and belief propagation, we propose the following algorithm for propagating element-wise means and variances of hidden features, along with their sensitivities, while simultaneously computing the gradient of the loss with respect to the weight parameters.

---

**Algorithm 1: Training BNN with backpropagation**

---

**Input**: Gaussian linear Bayesian neural network (Eqs. 3, 4).

**Output**: mean parameters $\hat{\mathbf{x}}_l$ and $\mathbf{P}_l$, their sensitivities $\nabla_{\hat{\mathbf{x}}_l} J$ and $\nabla_{\mathbf{P}_l} J$, gradient over weight distribution parameters $\nabla_{\boldsymbol{\theta}_{\mathbf{W}_l}} J$.

**Forward propagation**: For $l = 1, \ldots, L$, $\hat{\mathbf{x}}_l = \mathbf{f}(\hat{\mathbf{x}}_{l-1}, \boldsymbol{\theta}_{\mathbf{W}_l}, \boldsymbol{\omega}_l = 0)$, $\mathbf{P}_l = \hat{\mathbf{f}}'^2_{\mathbf{x}_{l-1}} \cdot \mathbf{P}_{l-1} + \hat{\mathbf{f}}'^2_{\boldsymbol{\omega}_l} \cdot \mathbf{1}$.

**Backpropagation**: For $l = L, \ldots, 1$, $\hat{\mathbf{x}}_{l|L} - \hat{\mathbf{x}}_l = G_l(\hat{\mathbf{x}}_{l+1|L} - \hat{\mathbf{x}}_{l+1})$, $\mathbf{P}_{l|L} - \mathbf{P}_l = G_l^2(\mathbf{P}_{l+1|L} - \mathbf{P}_{l+1})$.

**Gradient**: Eq. 8.

$\hat{\mathbf{x}}_{L|L} = \hat{\mathbf{x}}_L + P_L\left(\hat{\mathbf{g}}'^\top_{\mathbf{x}_L} \cdot \frac{\mathbf{y} - \mathbf{g}(\hat{\mathbf{x}}_L, 0)}{S_L}\right)$, $\quad \mathbf{P}_{L|L} = \mathbf{P}_L - \mathbf{P}_L^2\left((\hat{\mathbf{g}}'^2_{\mathbf{x}_L})^\top \cdot \frac{1}{\mathbf{S}_L}\right)$, $\quad \mathbf{S}_L = \hat{\mathbf{g}}'^2_{\mathbf{x}_L} \cdot \mathbf{P}_L + \hat{\mathbf{g}}'^2_{\boldsymbol{\nu}_L} \cdot \mathbf{1}$, and $\quad \mathbf{G}_l = \hat{\mathbf{f}}'^\top_{\mathbf{x}_l}(\mathbf{P}_l \cdot \mathbf{P}_{l+1}^{-\top})$. "$\cdot$" is matrix multiplication. $\mathbf{P}$ and $\bullet^2$ are element-wise. $\mathbf{f}$, $\hat{\mathbf{x}}_l$, $\mathbf{P}_l$, $\hat{\mathbf{f}}'_{\mathbf{x}_l}$, $\hat{\mathbf{f}}'_{\boldsymbol{\omega}_l}$, $\hat{\mathbf{g}}'_{\mathbf{x}_L}$, $\hat{\mathbf{g}}'_{\boldsymbol{\nu}_L}$ are defined in Eqs. 3 and 4. $\mathbf{1}$ is a 1-vector.

---

In a computational graph defined as $\mathbf{x}_0 = \mathbf{x}$, $\mathbf{a}_l = \mathbf{W}_l \cdot \mathbf{x}_{l-1}$, and $\mathbf{x}_l = \mathbf{f}_l(\mathbf{a}_l)$ for layers $l = 1, \ldots, L$, along with a mean-field Gaussian variational posterior for synaptic weights $\mathbf{W}_l \sim \mathcal{N}(\boldsymbol{\mu}_{\mathbf{W}_l}, \boldsymbol{\sigma}^2_{\mathbf{W}_l})$, implementing Algorithm 1 for Bayesian inference under these conditions is straightforward. The process involves propagating the activation mean ($\hat{\mathbf{x}}_{l-1}$) through the layer transformation ($\mathbf{f}_l$) once to compute $\hat{\mathbf{x}}_l$. Subsequently, activation variance ($P_{l-1}$) is propagated through these layer functions twice: first as $\mathbf{f}_l(\boldsymbol{\mu}^2_{\mathbf{W}_l} \cdot P_{l-1})$ and then as $\mathbf{f}_l(\boldsymbol{\sigma}^2_{\mathbf{W}_l} \cdot \hat{\mathbf{x}}^2_{l-1})$. The final variance ($\mathbf{P}_l$) is computed using element-wise multiplications and a smooth max (log-sum-exp) operation within the max-pooling process.

This algorithm is highly memory-efficient, involving element-wise gradient computations ($\hat{\mathbf{f}}'_{\mathbf{x}_l}$, $\hat{\mathbf{f}}'_{\boldsymbol{\omega}_l}$, $\hat{\mathbf{g}}'_{\mathbf{x}_L}$, $\hat{\mathbf{g}}'_{\boldsymbol{\nu}_L}$), as well as matrix-vector multiplications with a smoothing gain ($G_l$) to facilitate gradient descent with second-order information. In addition to updating synaptic weights, this algorithm simultaneously manages and updates weight variance on an element-wise basis. The power of automatic differentiation comes into play, streamlining the entire process. All that's required is specifying $\mathbf{P}_l$, a task that can also be automated to further enhance efficiency.

## 4  EXPERIMENTS

This section highlights belief propagation's effectiveness in training Bayesian neural networks for image classification, showing their competitive performance without normalization. We also evaluate UCI regression datasets and use visualization to understand model behavior. Our code demonstrates symbolic layers for probability distributions and gradients, revealing equivalence between Bayesian neural networks and graphical models. This expands auto-differentiation in TensorFlow, PyTorch, and JAX to support graphical models and variational inference techniques.

### 4.1  CLASSIFICATION

We demonstrate the effectiveness of the belief propagation algorithm in training various neural network architectures for image classification, considering them as Bayesian neural networks. Particularly, we show that incorporating natural gradient descent significantly improves generalization, eliminating the need for normalization techniques to stabilize and expedite training. Our experiments are designed for efficient one-day training cycles using Google Colab V100/A100. We utilize datasets including CIFAR 10/100 (Krizhevsky, 2009) and Tiny ImageNet (Le & Yang, 2015), along with architectural choices such as DenseNet-BC (Huang et al., 2017), ResNet (He et al., 2016), WideResNet-28-10 (Zagoruyko & Komodakis, 2016), EfficientNet B0 (Tan & Le, 2019), and MLP Mixer-S (Tolstikhin et al., 2021). Our training procedure adopts state-of-the-art practices, featuring a cosine annealing learning rate schedule (Loshchilov & Hutter, 2016) and image augmentation methods (Yun et al., 2019; Zhang et al., 2018b; Cubuk et al., 2020). We compare different configurations, including SGD with normalization layers (`vanilla NN`), Bayes-by-backprop (Blundell et al., 2015) with normalization layers (`reparam.`), Bayes-by-backprop with multiple neural networks per mini-batch without batch normalization (`reparam./NF+NG+ensemble`), belief propagation according to Algorithm 1 without batch normalization (`BP/NF`), and belief propagation with natural gradient without normalization (`BP/NF+NG`).

Table 1 presents the test errors. We observe significant performance improvements with Bayesian learning, particularly in less-regularized architectures. Natural gradient descent also enhances performance (`BP/NF` vs. `BP/NF +NG`). Additionally, Bayes-by-backprop, combined with an ensemble of neural networks and natural gradient descent, achieves competitive results, highlighting the variational aspect of normalization. Notably, we find that Bayesian learning without weight parameter sampling (Algorithm 1) exhibits faster convergence compared to methods involving weight sampling.

To assess the impact of weight randomness and sample randomness on Bayesian neural network (BNN) convergence and generalization, we analyzed empirical gradient variance with varying batch sizes and the evolution of training/validation loss over epochs. We conducted these analyses using MNIST and CIFAR 10 datasets, training a BNN with densely connected layers and a VGG16 model following the approach of (Wen et al., 2018). Our comparisons included our algorithm (`BP`) against reparameterization (`R`) (Blundell et al., 2015), local reparameterization (`L`) (Kingma et al., 2015), flipout (`F`) (Wen et al., 2018), and an ideal scenario with noise solely from mini-batching (`vanilla`). We observed that weight-induced noise in BNNs is substantial compared to mini-batch noise, and Bayesian learning with Algorithm 1 consistently achieved faster convergence than weight-sampling and non-Bayesian methods (Fig. 1a,c,b,d).

### 4.2  REGRESSION

We utilized UCI regression datasets to assess the performance of Bayesian neural network learning algorithms, aligning our experimental setup with the framework outlined in Hernández-Lobato

Table 1: A Bayesian neural network trained with belief propagation and natural gradient descent (BP/NF + NG), along with Bayes-by-backprop using an ensemble of neural networks and natural gradient descent (reparam./NF + NG + ensemble), achieves competitive image classification performance without normalization on state-of-the-art architectures.

| | | vanilla NN | reparam. | reparam./NF +NG+ensemble | BP/NF | BP/NF+NG |
|---|---|---|---|---|---|---|
| **CIFAR10** | DenseNet-BC | 4.51 | 4.75 | 4.50 | 4.40 | 4.35 |
| | ResNet | 5.46 | 5.70 | 4.75 | 4.35 | 5.10 |
| | WRN-28-10 | 3.50 | 3.65 | 3.50 | 3.40 | 3.30 |
| | Eff.NetB0 | 4.40 | 5.15 | 4.35 | 4.10 | 3.90 |
| | MLP Mixer-S | 8.20 | 9.35 | 8.55 | 6.70 | 4.85 |
| **CIFAR100** | DenseNet-BC | 22.27 | 23.05 | 21.05 | 21.90 | 20.50 |
| | ResNet | 27.22 | 29.55 | 28.35 | 27.10 | 25.90 |
| | WRN-28-10 | 18.80 | 18.95 | 18.90 | 18.30 | 17.70 |
| | Eff.NetB0 | 20.50 | 21.75 | 20.30 | 20.05 | 19.30 |
| | MLP Mixer-S | 30.60 | 32.90 | 32.25 | 28.35 | 26.00 |
| **Tiny ImageNet** | DenseNet-BC | 30.10 | 30.50 | 30.15 | 29.95 | 29.25 |
| | ResNet | 31.50 | 33.00 | 32.00 | 31.50 | 30.70 |
| | WRN-28-10 | 28.70 | 29.15 | 28.75 | 28.55 | 27.30 |
| | Eff.NetB0 | 29.45 | 30.00 | 29.25 | 29.10 | 28.95 |
| | MLP Mixer-S | 35.85 | 38.45 | 36.35 | 34.30 | 31.10 |

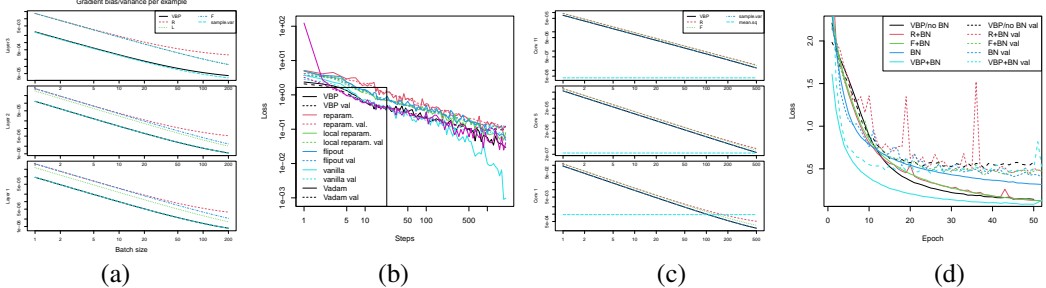

(a)      (b)      (c)      (d)

Figure 1: Gradient Variance vs Batch Size for Dense and VGG16 Networks (a and c) and Learning Algorithm Convergence (b and d). Bayesian Neural Formulation Enhances Convergence.

& Adams (2015). Our model (Algorithm 1) is compared against a deep ensemble model (Lakshminarayanan et al., 2017) and the local parameterization algorithm (Kingma et al., 2015). Table 2 presents a comparison of the three approaches based on root mean square error and log probability.

Table 2: Log Probability and RMSE of Belief Propagation, Deep Ensemble, and Local Reparameterization on UCI Data.

| | BP | | DE | | Reparam | |
|---|---|---|---|---|---|---|
| | logprob | rmse | logprob | rmse | logprob | rmse |
| boston | $-2.6 \pm 0.4$ | $3.5 \pm 1.0$ | $-2.5 \pm 0.1$ | $3.3 \pm 1.0$ | $-3.2 \pm 0.7$ | $3.5 \pm 1.1$ |
| concrete | $-3.2 \pm 0.3$ | $6.1 \pm 0.8$ | $-3.4 \pm 0.1$ | $7.7 \pm 0.6$ | $-3.9 \pm 0.2$ | $6.8 \pm 0.5$ |
| energy | $-2.2 \pm 0.2$ | $2.7 \pm 0.4$ | $-2.2 \pm 0.1$ | $2.7 \pm 0.3$ | $-3.0 \pm 0.3$ | $2.9 \pm 0.3$ |
| kin8nm | $1.1 \pm 0.0$ | $0.1 \pm 0.0$ | $0.4 \pm 0.0$ | $0.2 \pm 0.0$ | $-2.3 \pm 0.5$ | $0.1 \pm 0.0$ |
| naval | $4.0 \pm 2.1$ | $0.0 \pm 0.0$ | $2.9 \pm 0.1$ | $0.0 \pm 0.0$ | $4.9 \pm 0.1$ | $0.0 \pm 0.0$ |
| power | $-2.8 \pm 0.0$ | $4.1 \pm 0.2$ | $-3.0 \pm 0.0$ | $4.3 \pm 0.2$ | $-3.2 \pm 0.1$ | $4.2 \pm 0.2$ |
| protein | $-2.9 \pm 0.0$ | $4.4 \pm 0.1$ | $-3.0 \pm 0.0$ | $5.1 \pm 0.1$ | $-3.5 \pm 0.1$ | $4.5 \pm 0.0$ |
| wine | $-1.0 \pm 0.1$ | $0.6 \pm 0.0$ | $-0.9 \pm 0.1$ | $0.6 \pm 0.0$ | $-1.4 \pm 0.2$ | $0.6 \pm 0.0$ |
| yacht | $-2.8 \pm 0.4$ | $64 + 19$ | $-2.7 + 0.2$ | $59 + 17$ | $-2.5 + 0.3$ | $4.1 + 0.9$ |

Different from previous works, we also visualize the observation-label distribution of a Bayesian neural network through Monte Carlo simulations and 2D projections (Fig. 2). Starting with slight perturbations to the training data, we employ Hamiltonian Monte Carlo (Neal et al., 2011) to target uncalibrated probabilities (Eq. 2) and monitor convergence with the Gelman-Rubin diagnostic. This produces simulated feature-label patterns representing typical model observations and loss.

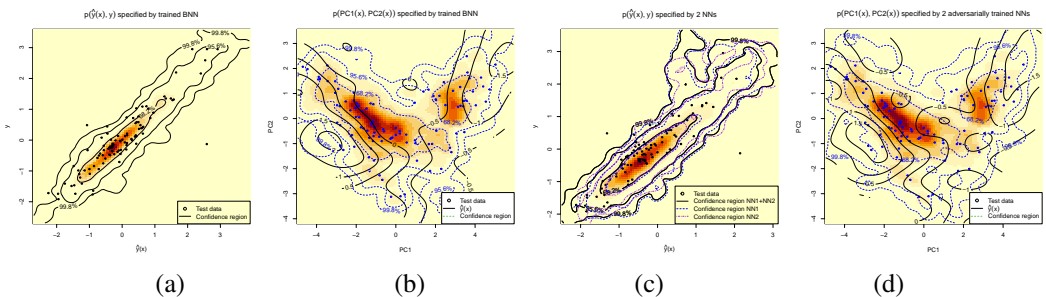

Figure 2: Joint Distributions from Neural Network Ensembles on Boston Housing Data. Projections: Predicted vs. True Labels (a, c) and Principal Components of Largest Variance (b, d). (a, b) Bayesian NN (Algorithm 1), (c, d) Adversarial NNs.

In Fig. 2 (a) and (c), we project this distribution onto predicted vs. true labels. Ideal models concentrate probability on the diagonal, indicating accurate predictions, especially for unfamiliar cases. Bayesian neural networks trained with Algorithm 1 tightly cluster observations around the diagonal (Fig. 2 (a)), whereas independently trained networks diverge off-diagonal due to limited data (Fig. 2 (c)). Ensembles produce predictions closer to the diagonal. In Fig. 2 (b) and (d), we project observations onto the two principal components, outlining confidence regions and marking predictions. Bayesian neural networks focus observations where predictions incur less loss (Fig. 2 (b)), while non-Bayesian networks have scattered, less smooth expectations. These insights contribute to understanding model behavior, robustness, and susceptibility to adversarial attacks.

## 5  RELATED WORKS

This paper presents a novel perspective on Bayesian neural networks (BNNs) by establishing their equivalence to Markov random fields and linking backpropagation with belief propagation. It distinguishes itself from prior work, which mainly focused on deriving learning algorithms via MCMC and variational inference, using BNNs in NLP and computer vision, and formulating Bayesian learning biases. Bayesian deep learning encompasses both deterministic and MCMC-based methods, including Hamiltonian Monte Carlo (Neal et al., 2011), Stochastic Gradient Langevin Dynamics (Welling & Teh, 2011), and stochastic weight averaging (Maddox et al., 2019). Deterministic techniques like Bayes by backpropagation (Blundell et al., 2015), Laplace approximation (MacKay, 1992; Barber & Bishop, 1998), and natural gradient approximations (Zhang et al., 2018a; Khan et al., 2018) coexist with deep learning-specific methods such as probabilistic backpropagation (Hernández-Lobato & Adams, 2015), deep ensembles (Lakshminarayanan et al., 2017), dropout (Gal & Ghahramani, 2016), and variance-reduction enhancements (Kingma et al., 2015). The revealed equivalence between BNNs and Markov random fields enables the sharing of learning algorithms. BNNs are also versatile, capturing diverse deep learning inductive biases, including adversarial learning (Guo et al., 2017; Ye & Zhu, 2018), semi-supervised learning (Gordon & Hernández-Lobato, 2020), meta-learning (Ravi & Beatson, 2018; Yoon et al., 2018), transfer learning (Maddox et al., 2019), and continual learning (Pan et al., 2020; Daxberger et al., 2021), with robustness against adversarial attacks (Uchendu et al., 2021; Pang et al., 2021). Applications span natural language processing (Shi et al., 2020; Yu et al., 2022), computer vision (Wang et al., 2017), graph learning, time series analysis, and reinforcement learning.

## 6  CONCLUSIONS

Our research illuminates the noteworthy parallels between probabilistic graphical models and Bayesian neural networks, highlighting their synergistic roles in theoretical and practical realms. These similarities foster a reciprocal enhancement of methodologies across both fields. Our empirical validations across diverse scenarios underscore the effectiveness of our approach. This work presents a novel perspective on Bayesian Neural Networks and probabilistic graphical models, fostering advancements in model development and enriching our understanding of their interrelation.

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
