# OpenReview forum: "Duality of Information Flow: Insights in Graphical Models and Neural Networks"
_ICLR.cc/2024/Conference — Submitted to ICLR 2024_

### Official Review · Reviewer_Ac7q · 2023-10-28

**Soundness:** 3 good
**Presentation:** 2 fair
**Contribution:** 3 good
**Rating:** 6
**Confidence:** 2

**Summary:**

This paper focuses on the relationship between the graphical models and the neural networks. The authors first indicate the equivalence between the belief propagation and the back propagation. Then the duality between the Bayesian neural networks and the graphical models are specified via the relationship between Langevin and the Fokker-Planck dynamics. Based on these observations, the authors propose a new training method, whose efficacy is demonstrated by the numerical results.

**Strengths:**

This paper provides both theoretical analysis and the empirical verification for the relationship between the Bayesian neural networks and the Markov random fields. The results of this paper are interesting for the deep learning community, and they can serve the basis for understanding the training process of neural networks.

**Weaknesses:**

1. The theoretical results of this paper are built on the basis of Gaussian linear model. This can be strict in many realistic problems. It would be helpful to discuss the extension of this model. For example, is it possible to extend the results in this paper to the mixture of Gaussian.

2. The writing of the theoretical results can be improved. The authors define many things in the Theorems 1 and 2. It will be more friendly to readers to define these quantities before the statement of the theorems.  In addition, the fontsize of the math characters should be consistent.  For example, in (3) of Theorem 1, the fontsize of characters changes after $=$.

**Questions:**

The questions are listed in the Weakness part.

---

> ### Author Response · Authors · 2023-11-18
> **reformatting theorems for reader-friendliness & on the limitations of Gaussian linear model**
>
> Thank you for your valuable reivew!
>
> **Concerning reformatting theorems for reader-friendliness:** In our revised manuscript, we have repositioned the key quantities and concepts to precede the statements of the theorems, along with concise explanations for each. This reorganization aims to enhance clarity and comprehension for our readers. Additionally, we have included brief, intuitive explanations of the central results within each theorem. The revised sections now reads as follows:
>
> *Theorem 1: Equivalence between Backpropagation and Belief Propagation in a Bayesian Neural Network.*
>
> (1) *Gradient of Loss with Respect to Parameters:* In its most general form, the gradient of loss with respect to the post-activation filtering distribution parameters $\boldsymbol{\theta}\_{l}$ and the variational weights distribution parameters $\boldsymbol{\theta}\_{\mathbf{W}\_{l}}$ can be expressed as:
>
> $\nabla_{\boldsymbol{\theta}\_{l}} J=\dots$, $\nabla_{\boldsymbol{\theta}\_{\mathbf{W}\_{l}}} J =\dots$.
>
> (2) *Error Gradient for Canonical Parameters:* When filtering distribution is in the exponential family, error gradient for the canonical parameters is the difference in the mean parameters between smoothing and filtering distributions. Backpropagation mirrors backward belief propagation:
>
> $\nabla_{\boldsymbol{\theta}\_{l}} J=\dots$, $\nabla_{\boldsymbol{\theta}\_{\mathbf{W}\_{l}}} J =\dots$.
>
> (3) *Error Gradient for Post-Activation Mean and Variance:* In a Gaussian linear Bayesian neural network, the error gradient for post-activation mean and variance is related to the difference between smoothing and filtering mean and variance in the corresponding Gaussian linear dynamics. Gradient backpropagation parallels belief backward propagation induced by smoothing gain $G_l$ and $G_{\mathbf{W}_l}$:
>
> $\nabla_{\hat{\mathbf{x}}\_{l}}J = \dots$, $\nabla_{P\_{l}}J = \dots$, $\nabla_{\hat{\mathbf{W}}\_{l}}J = \dots$, $\nabla_{\boldsymbol\sigma_{\mathbf{W}_l}^2}J =\dots$.
>
> The error gradient for the canonical parameters of post-activation filter distributions in a Gaussian linear Bayesian neural network is the difference in the first and second moments between smoothing and filtering distributions in the corresponding Gaussian linear dynamics. Gradient backpropagation again parallels belief backward propagation induced by smoothing gain:
>
> $\nabla_{\boldsymbol{\eta}\_{l}}J =\dots$, $\nabla_{\boldsymbol{\Lambda}\_{l}}J =\dots$
>
> *Theorem 2: Relationship between the Langevin and the Fokker-Planck Dynamics of a Bayesian Neural Network.*
>
> (1) For post-activations distributed as per an exponential family, the error gradient relative to post-activation equals the potential energy gradient, resulting from differences in canonical parameters between filtering and smoothing distributions:
>
> $\nabla_{\mathbf{x}\_{l}}\log  p(\mathbf{y}|\mathbf{x}\_{l},\mathbf{x}\_{0}=\mathbf{x}) =\dots$.
>
> This difference in canonical parameters defines the orthogonal projection from the Jacobian of the sufficient statistics to the error gradient relative to post-activation. Here, $\bullet^+$ is the pseudo-inverse.
>
> $\boldsymbol{\theta}\_{l}-\boldsymbol{\theta\_{l|L}} =\dots$
>
> (2) In Gaussian-linear Bayesian neural networks, drift and diffusion processes, defined by the error gradient and Hessian, guide post-activations toward the variational posterior of Gaussian linear dynamics as follows:
>
> $\nabla_{\mathbf{x}\_{l}}\log p(\mathbf{y}|\mathbf{x}\_{l},\mathbf{x}\_{0}=\mathbf{x})=\dots$,
> $\nabla_{\mathbf{x}\_{l}\mathbf{x}\_{l}^{\top}}\log p(\mathbf{y}|\mathbf{x}\_{l},\mathbf{x}\_{0})=\dots$,
> $\nabla_{\hat{\mathbf{x}}\_{l}}\log p(\mathbf{y}|\mathbf{x}\_{0})=\dots$,
> $\nabla_{P_{l}}\log p(\mathbf{y}|\mathbf{x}\_{0}=\mathbf{x}) =\dots$.
>
> (3) When the weight variance vanishes, making the state transition deterministic with  $p(\mathbf{x}\_{l+1}|\mathbf{x}\_{l})=\delta(\mathbf{x}\_{l+1}-f(\mathbf{x}\_{l}))$, Bayesian back propagation degenerates into non-Bayesian back propagation:
>
> $\nabla_{\mathbf{x}\_{l}}\log p(\mathbf{y}|\mathbf{x}\_{l})=$
>
> **Concerning the limitations of Gaussian linear model:** Thank you for your valuable input. Our research establishes a fundamental framework, accommodating a diverse array of weight variational posteriors and inference methods. While our algorithm is rooted in Gaussian linear models, it also aligns with the approximate linearity of neural networks in overparameterized regimes, as detailed by Jacot et al. (2018) in their study on neural tangent kernels. This alignment is substantiated by our empirical findings, showcasing effective performance across practical scenarios. We appreciate your recommendation to expand our model to include mixtures of Gaussians, recognizing the potential to handle more intricate data distributions.

---

> > ### Comment · Reviewer_Ac7q · 2023-11-23
> >
> > Thank the authors for the detailed response! The response addressed my concerns. It is encouraged to include the discussion about the extension beyond the Gaussian linear case in the main paper. I will maintain my scores.

---

### Official Review · Reviewer_2p2Z · 2023-11-01

**Soundness:** 2 fair
**Presentation:** 2 fair
**Contribution:** 2 fair
**Rating:** 3
**Confidence:** 3

**Summary:**

The paper investigates the parallels between probabilistic graphical models and Bayesian neural networks, specifically the main results highlights the equivalence between message passing in probabilistic graphical models and belief propagation in neural networks. Through empirical assessments conducted across diverse scenarios, the paper substantiates this convergence, highlighting their equivalence

**Strengths:**

One of the primary strengths of the paper lies in its problem statement which is exploratory in nature. Specifically, the paper seeks to establish an equivalence between Bayesian neural networks and Markov random fields.

**Weaknesses:**

- While the exploratory nature of the problem is acknowledged, the paper could benefit from a stronger motivation. It is essential to explain why the question of equivalence between Bayesian neural networks and Markov random fields is important. Clarifying the practical implications and real-world significance of this equivalence would strengthen the paper's rationale.
- A brief note on potential future directions would enhance the appeal of the paper.

**Questions:**

- The paper asserts an equivalence between Bayesian neural networks and probabilistic graphical models, suggesting the potential for "enhanced models." However, the term "enhanced models" remains vague and requires precise definition. Providing insight into how this equivalence can be leveraged to create more effective or advanced models would enhance the paper's clarity.

-  The equations in the paper are dense and lack organization, making the paper challenging to comprehend. To improve readability, the authors should offer intuitive explanations for their main results and their consequences. Additionally, providing a brief sketch of the proofs, especially in Theorems 1 and 2, which establish the equivalence between back-propagation and belief propagation, would be beneficial. Clearly defining terms like "tensor particles," "stochastic tensor flow," "variational message passing," "sensitivity of probability distribution," "tensor distribution evolution," and "mean field Gaussian variational posterior" is crucial to ensure clarity and readability.

- The plots in Figure 1 are too small to discern the axes, scaling, and legend effectively. The authors should consider enlarging the plots or providing clearer visual aids to improve the readability of the figures.

-  Figure 2's main point is unclear, and it is not evident why the plots emphasize the diagonal. The authors should provide a more detailed explanation of the purpose and focus of Figure 2 to enhance the reader's understanding.

- The paper's GitHub link for the code is not anonymized, compromising the authors' anonymity and violating the double-blind review protocol. The authors should address this issue promptly to maintain the integrity of the double-blind review process.

---

> ### Author Response · Authors · 2023-11-18
> **balancing contents within the 9-page limit, restatement of motivation, significance, and potential future directions**
>
> Thank you for your valuable reivew!
>
> **Concerning authors' anonymity and violation of the double-blind review protocol:** Thank you for your concern. We certify that there is no URL (e.g., GitHub page) that could be used to find our identity. We are perhaps not who you guessed we are.
>
> **Concerning providing a brief sketch of the proofs:** Thank you for your suggestion regarding the presentation of our proofs. We did. The complete proofs involve complex concepts, such as convex conjugacy and matrix derivatives, so they are detailed in the supplementary material. We have included brief, intuitive explanations of the central results within each theorem in the revised manuscript.
>
> **Concerning clearly defining terms:**
>
> - "tensor *particles*," "tensor distribution evolution," pp 6 and chapters on Fokker-Planck equations and Stochastic Processes in Gardiner et al., (1985)
> - "*sensitivity* of probability distribution", Eq. 14 (pp 291) in Stork et al., (2000)
> - "variational message passing" Section 3.3, Definition of the Variational Message Passing algorithm in Winn et al., (2005)
> - "mean field Gaussian variational posterior", Examples 5.2 & 5.3, Chapter 5, Wainwright & Jordan, (2008)
>
> **Concerning motivation, significance, and implications:**
>
> *Motivation*
>
> Our primary motivation is to advance the theoretical understanding of deep learning, focusing on the general assumptions necessary for learning to generalize and converge. Our approach is inspired by the "society of mind" concept, which proposes that intelligence emerges from the interactions of simple agents, similar to various networks processing information and optimizing operations. Our research aligns with a lineage that views deep learning through the lens of probability theory, encompassing Bayesian neural networks, Boltzmann machines, variational and Monte Carlo methods, deep generative models, and neural network Gaussian processes.
>
> *Significance*
>
> Our research, highlighting the parallels between graphical models and neural networks, is significantly practical and theoretical. It enables learning and inference within graphical models using auto-differentiation in mainstream deep learning APIs like TensorFlow, PyTorch, and JAX. Our code demonstrates the creation of symbolic layers for handling probability distributions and error gradients, seamlessly integrating with standard non-Bayesian layers. This advancement broadens the spectrum of deep neural network architectures and learning algorithms, incorporating a diverse range of graphical models and variational inference methods.
>
> Theoretically, our work lays down a mathematical foundation to discuss the probability of choosing an effective non-Bayesian neural network from a Bayesian neural network ensemble based on a specific training dataset, in line with statistical learning theory. Additionally, our introduction of the Gaussian linear neural Gaussian Markov random field model establishes parameters for global smoothness and local continuity in high-dimensional Gaussian distributions, aiding in the calculation of stochastic gradient descent convergence rates. This is particularly relevant given the tendency of neural networks to maintain approximate linearity in overparameterized settings.
>
> *Future Directions:*
>
> - Exploring normalization-free neural networks for SOTA performance in computer vision and NLP tasks.
> - Developing Bayesian language models, including large hierarchical hidden Markov models, to compete with transformer architectures.
> - Investigating graphical models that mirror transformer functionality and exploring improvements for better performance with fewer parameters.
> - Utilizing Fisher divergence within the Bayesian neural network framework to advance deep generative models, possibly offering alternatives to sliced score matching.
> - Establishing generalization error bounds using concentration inequalities and demonstrating these theoretical results with SOTA architectures.
> - Examining the convergence rate of learning SOTA neural networks using neural Gaussian linear Markov random field approximations.
> - Evolving neural network architectures from data, exploring structural features of complex systems, and evolving neural architectures from scratch.
> - Exploring the equivalence between intelligence and randomness, establishing a bidirectional relationship between data and synaptic weight distributions, and investigating the functional equivalence of synaptic weights.
>
> **Figure 1 are too small:** We have enlarged the panels as much as we can to not exceed the 9-page limit.
>
> **Figure 2's main point:** Main point is that Markov random field is a generative model. We added detailed explanations in the supplementary material in the revised manuscript.

---

### Official Review · Reviewer_kosy · 2023-11-09

**Soundness:** 3 good
**Presentation:** 2 fair
**Contribution:** 3 good
**Rating:** 5
**Confidence:** 3

**Summary:**

This paper identifies and establishes connections between Bayesian neural networks and probabilistic graphical models (Markov random fields). In particular, they establish equivalence between backpropagation and belief propagation in Theorem 1. Their Theorem 2 presents a relationship between Langevin and the Fokker-Planck dynamics. They further leverage these connections to develop a belief propagation-based algorithm to train Bayesian neural networks and show its efficacy through numerical experiments.

**Strengths:**

The connection between backpropagation and belief propagation is an interesting result. It has the potential to pave the way for the adaptation of various algorithms from the field of probabilistic graphical models to Bayesian neural networks. The authors have also substantiated their theory through a series of numerical experiments, demonstrating that their proposed algorithm consistently surpasses other baseline methods in terms of performance.

**Weaknesses:**

The paper is quite heavy on notations and a little difficult to follow at times. Some of the notations are inconsistent and create confusion while reading. Few examples (there could be more):
1. I believe it would be easier to understand the equations if the dimensions of the parameters were stated clearly.
2. $f_l$ seems to take arguments in different orders (check eq. for $x_l$ before (4) and then in the fifth line after (5)).
3. Is there a purpose for italicizing one of the $f_{x_{l-1}}$ in (6)?
4. In Theorem 1 (2), $T(x_l)$ is written in both regular and bold fonts. In general, there does not seem to be a consistent interpretation of boldface parameters.

Check the following statement (and other similar statements): "This research emphasizes the convergence between probabilistic graphical models and neural networks, revealing their intrinsic parallels."
It seems that the authors are claiming that probabilistic graphical models and neural networks are the same. I understand the parallels but as a whole the statement seems to be too strong. It would be better to qualify such string statements by formal evidence.

Please also see the questions section.

**Questions:**

1. The term tensor "particles" has been used repeatedly throughout the paper without a formal definition or reference. Could you please explain what exactly is a tensor particle?
2. In the context of probabilistic graphical models, belief propagation comes with no convergence guarantees for graphs with cycles. Does such consideration occur in the proposed method? In general, would the proposed algorithm always converge?
3. Could you explain the following comment under Theorem 1 (a formal derivation or a pointer to a reference would help)?
"The exponential family assumption in (2) generally holds as long as the joint probability density of the random variables is strictly positive, as per the Hammersley-Clifford theorem (Hammersley & Clifford, 1971)."
4. I am trying to place this work in the context of the existing work relating belief propagation with neural networks. Has there been any work in this regard or are the authors presenting a completely novel observation?

I am open to changing my score based on the answers from the authors.

---

> ### Author Response · Authors · 2023-11-18
> **authors presenting a completely novel observation & revisions for user-friendliness**
>
> Thank you for your valuable reivew!
>
> **Regarding dimensions in our equations**. In our notation, operators are applied from the left, meaning,
> $\mathbf{W}\_{l}\cdot \mathbf{x}\_{l-1}=\left( \sum_{j_{l-1}}\textnormal{W}\_{i_l,j_{l-1}}\textnormal{x}\_{j_{l-1}} \right)\_{i_l}$
> calculates pre-activation elements at multi-index $i_l$ as weighted sums of post-activation elements at $j\_{l-1}$. For ease of understanding, multi-indexes can be viewed as standard integer indexes, with synaptic weights as matrices and post-activation tensors as vectors. This clarification is in the revised paper's first paragraph, Section 2, Notation.
>
> **Regarding notation consistency**: We use `\mathbf` for synaptic weights, post-activations, inputs, and outputs; `\boldsymbol` for weight and post-activation distribution parameters; and standard symbols for other mathematical elements. This notation has been thoroughly checked for consistency in the revised paper.
>
> **Responding to the overstatement concern**: We aim to highlight similarities between probabilistic graphical models and neural networks, not asserting their identity but their ability to inform and enhance each other's methodologies. We have amended the statement to better reflect this nuanced perspective in the revised paper.
>
> **On the term "tensor particles"**: It describes a set comprising inputs, post-activations, synaptic weights, and labels in a Bayesian neural network, drawing from Monte Carlo methods in statistical physics, where a "particle" represents a possible state of the system under study, similar to how physical particles behave and interact (Gardiner et al., 1985). This explanation is added at the end of Section 1 Introduction's 2nd paragraph.
>
> **Addressing convergence in belief propagation**: While it's true that the exact conditions for loopy belief propagation's convergence are not fully established, our theoretical contributions—specifically the equivalence between backpropagation and belief propagation in Bayesian neural networks, and the analysis of Langevin and Fokker-Planck dynamics—are foundational. They enable the use of other approximate methods with known convergence guarantees, such as variational and Monte Carlo methods. Our approach, centered on Gaussian belief propagation, incorporates well-understood convergence conditions that are easier to analyze. Practically, our algorithm shows faster convergence compared to stochastic gradient descent for non-Bayesian networks and Monte Carlo methods for Bayesian neural networks, as demonstrated in Figure 1 and detailed in Section 4.1.
>
> **Clarifying Theorem 1's exponential family assumption**: The Hammersley–Clifford theorem states that a strictly positive probability measure Markov with respect to a graph G equates to a Gibbs random field. Our theorem's Bayesian neural network computational graph forms an undirected graphical model, satisfying this and ensuring a Gibbs measure. The proof of the Hammersley-Clifford Theorem is technical; a machine learning-aligned version is under Theorem 7.12 in Grimmett's "Probability on Graphs." It is based on establishing the equivalence between global Markov property, local Markov property, and pairwise Markov property. To prove factorization from pairwise Markov property, they show that that the potential on any non-complete subgraph $a$ is 0. This is translated into $\phi_a(x)=\sum_{b \subseteq a} (-1)^{|a\backslash b|} \log f_b\left(x_b, x_{b^c}^\star\right)=0$. The proof of this, in turn, is based on Mobius Inversion Lemma.
>
> **Regarding novel contributions**: In this paper, we presented a **completely novel observation**, specifically: In the context of existing work on belief propagation and neural networks (cf. Section 5 Related Works), our paper introduces a unique angle by demonstrating the equivalence of Bayesian neural networks (BNNs) with Markov random fields, connecting backpropagation with belief propagation, and relating probabilistic perspective of tensor flow at the microscopic level and the deterministic perspective of tensor distribution evolution at the macroscopic level. Previous research primarily focused on developing learning algorithms through MCMC and variational inference, applying BNNs in NLP and computer vision, and exploring Bayesian learning biases. Extending beyond the scope of our surveyed works, if parallels existed in areas like neural tangent kernels, neural network Gaussian processes, or neural ordinary differential equations, they would likely entail interpreting neural networks through the lens of established mathematical/probability theories. However, our extensive review of the recent literature revealed no such works aligning closely with our specific methodologies and theoretical insights in these domains.

---

### Author Response · Authors · 2023-11-23
**Improving Reader-Friendliness, Novelty Confirmation, & Technical Details Discussion**

Dear Reviewers,

We sincerely appreciate your valuable feedback and constructive comments. Your insights have greatly contributed to improving the quality of our paper. Based on your feedback, we have addressed several key points, and here's a summary of our responses:

1. **Improving Reader-Friendliness**:
   - We reorganized the paper to enhance clarity and comprehension. Key quantities and concepts are now factored outside of the theorems, with brief explanations provided for each.
   - We clarified terminology and provided explanations, citing sources where necessary, to align with machine learning literature.
   - Notation has been thoroughly checked for consistency and clarified in Section 2 for better understanding.

2. **Novelty Confirmation**:
   - We emphasized the novelty of our work in Section 5 (Related Works) by highlighting that our research introduces completely novel observations. We also conducted an extensive literature review, finding no closely aligned works in the domains of neural tangent kernels, neural network Gaussian processes, or neural ordinary differential equations.

3. **Technical Details Discussion**:
   - Specific technical questions and concerns from individual reviewers have been addressed in our responses to their respective comments.

Our work not only presents novel theoretical insights but also has practical significance, enabling learning and inference within graphical models through auto-differentiation in mainstream deep learning frameworks. This is exemplified by our code, which showcases the practical utility of our research.

The revised version, together with the version with changes highlighted in blue in the "main+supplementary.pdf" supplementary material, have been uploaded for your review.

---

### Meta-Review · Area_Chair_48ot · 2023-12-04

**Metareview:**

I am familiar with the two topics discussed in this paper -- probabilistic graphical models and neural networks. This paper has a lot of potential. However, reviewers were generally rather critical about the density of the results and the numerous terminology that is introduced. Granted, the two areas are somewhat disparate but I do feel that the reviewers have a point, having gone through the paper in some detail. In particular, Theorem 1 is long and it is difficult to extract its implication. Given the connection between the training dynamics of neural networks and belief propagation, what does that say about the practical implications of training of neural networks? Is the linear Gaussian assumption too restrictive in practice? I believe these have to be answered in a definite manner for the paper to realize its true potential.

**Justification For Why Not Higher Score:**

There are several limitations as mentioned above.

**Justification For Why Not Lower Score:**

The score is already quite low.

---

### Decision · Program_Chairs · 2024-01-16

Reject